

# Low-energy excitations and transport functions of the one-dimensional Kondo insulator

Robert Peters[1*] and Roman Rausch[2]

**1** Department of Physics, Kyoto University, Kyoto 606-8502, Japan
**2** Technische Universität Braunschweig, Institut für Mathematische Physik,
Mendelssohnstraße 3, 38106 Braunschweig, Germany

⋆ peters@scphys.kyoto-u.ac.jp

## Abstract

Using variational matrix product states, we analyze the finite temperature behavior of a half-filled periodic Anderson model in one dimension, a prototypical model of a Kondo insulator. We present an extensive analysis of single-particle Green's functions, two-particle Green's functions, and transport functions creating a broad picture of the low-temperature properties. We confirm the existence of energetically low-lying spin excitations in this model and study their energy-momentum dispersion and temperature dependence. We demonstrate that charge-charge correlations at the Fermi energy exhibit a different temperature dependence than spin-spin correlations. While energetically low-lying spin excitations emerge approximately at the Kondo temperature, which exponentially depends on the interaction strength, charge correlations vanish already at high temperatures. Furthermore, we analyze the charge and thermal conductivity at finite temperatures by calculating the time-dependent current-current correlation functions. While both charge and thermal conductivity can be fitted for all interaction strengths by gapped systems with a renormalized band gap, the gap in the system describing the thermal conductivity is generally smaller than the system describing the charge conductivity. Thus, two-particle correlations affect the charge and heat conductivities in a different way resulting in a temperature region where the charge conductivity of this one-dimensional Kondo insulator is already decreasing while the heat conductivity is still increasing.

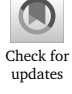

## 1    Introduction

Kondo insulators [1–3], such as $SmB_6$ [4,5] and $YB_{12}$ [6], are strongly correlated insulators, where the resistivity becomes large at low temperatures due to a gap in the single-particle spectrum. This gap arises at the Fermi energy due to the hybridization between itinerant conduction ($c$) electrons and strongly correlated $f$ electrons. Because of the correlation effects in the $f$ electrons, the temperature dependence of the resistivity of Kondo insulators follows a different behavior than that of uncorrelated band insulators: At high temperatures, these materials behave as metals due to itinerant conduction electrons. At these temperatures, the $f$ electrons are localized and do not affect electronic properties around the Fermi energy. When cooling down the material, the hybridization between localized $f$ electrons and $c$ electrons leads to the Kondo effect, resulting in a singlet formation between the $f$ electrons and $c$ electrons, which opens a single-particle gap at the Fermi energy. Because this single-particle gap can also be understood as a hybridization gap between the $c$-electron band and an $f$-electron band, Kondo insulators are sometimes described as band insulators, which has led to the insight that Kondo insulators can be distinguished into topologically trivial and nontrivial insulators according to their band structure [3,7]. The realization that topological Kondo insulators are strongly correlated topological insulators intensified the research on these materials leading to predictions of phenomena that cannot be observed in noninteracting topological insulators [8–13].

However, although the single-particle spectrum can be understood using band theory, it is long clear that Kondo insulators are more than mere band insulators. Analytical and numerical calculations in one dimension have shown low-lying spin excitations in a Kondo insulator [14–22]. It has been shown that the spin gap is exponentially small, much smaller than the single-particle gap. Thus, Kondo insulators possess excitations within the single-particle gap. While most of these calculations have been done for one-dimensional systems, quantum Monte Carlo calculations and cluster calculations have been performed in two dimensions, also demonstrating low-lying spin excitations [23–27].

Recently, new experiments on Kondo insulators $SmB_6$ [28,29] and $YbB_{12}$ [30] revealed unexpected results. Quantum oscillations in strong magnetic fields have been observed in these materials while insulating. Besides conventional theories based on magnetic breakdown [31–34], correlation effects [35], and surface conduction [36], observable quantum oscillations in

the insulating state have been hotly debated arising from charge-neutral particles forming a Fermi surface, such as Majorana fermions [37], and excitons [38, 39]. Furthermore, specific heat and thermal transport measurements on $YbB_{12}$ [40] and $YbIr_3Si_7$ [41] have revealed low-lying excitations in these Kondo insulators that can transport heat but no electric current. Thus, it has been argued that these itinerant charge-neutral excitations are responsible for thermal transport and lead to observable quantum oscillations in strong magnetic fields. However, a conclusion to these recent experiments is not yet found.

In this paper, we reexamine one-dimensional Kondo insulators using the highly accurate variational-matrix-product states (VMPS) technique at $T \geq 0$. We analyze the single-particle spectral function, charge-structure factor (CSF), and spin-structure factor (SSF) and their temperature dependencies. In particular, we confirm the existence of energetically low-lying spin excitations in the Kondo insulating state and study their energy-momentum dispersion and temperature dependence. We demonstrate that charge-charge correlations at the Fermi energy exhibit a very different temperature dependence than these spin-spin correlations. While energetically low-lying spin excitations emerge approximately at the Kondo temperature, charge correlations vanish already at high temperatures. Furthermore, we analyze the charge and thermal conductivity at finite temperatures by calculating the time-dependent current-current correlation functions. While both charge and thermal conductivity can be fitted for all interaction strengths by gapped systems with a renormalized band gap, the gap in the system describing the thermal conductivity is generally smaller than the system describing the charge conductivity. While the lowest temperature at which we can accurately calculate the conductivity is limited due to numerical constraints, which prevents our calculations from reaching the temperatures studied in the experiments, we can clearly conclude that there is a temperature region where the charge conductivity of this one-dimensional Kondo insulator is already decreasing while the heat conductivity is still increasing.

The rest of this paper is organized as follows: In Sec. 2, we describe the technical details of the model and the calculations. This is followed by a discussion on dynamical correlation functions at $T = 0$ and $T > 0$ in Sec. 3. In Sec. 4, we show the time-resolved current-current correlation functions and calculate the charge and thermal conductivity. Finally, in Sec. 5, we summarize and conclude the paper.

## 2 Model and Method

### 2.1 Model

To study the properties of Kondo insulators in one dimension (1D) with $L$ sites, we use the 1D periodic Anderson model, which reads

$$H = \sum_{k\sigma} \left( \epsilon_k^c c_{k\sigma}^\dagger c_{k\sigma} + \epsilon_k^f f_{k\sigma}^\dagger f_{k\sigma} \right) + V \sum_{j\sigma} \left( f_{j\sigma}^\dagger c_{j\sigma} + c_{j\sigma}^\dagger f_{j\sigma} \right) + \sum_j \left( U n_{j\uparrow}^f n_{j\downarrow}^f + E_f \left( n_{j\uparrow}^f + n_{j\downarrow}^f \right) \right),$$

$$\epsilon^c = -2t \cos(k),$$

$$\epsilon_f = 0,$$

(1)

where $c_{k\sigma}^\dagger$ ($f_{k\sigma}^\dagger$) creates a $c$ ($f$) electron with momentum $k$ and with spin projection $\sigma = \{\uparrow, \downarrow\}$. The particle densities on site $j$ are $n_{j\sigma}^f = f_{j\sigma}^\dagger f_{j\sigma}$ and $n_{j\sigma}^c = c_{j\sigma}^\dagger c_{j\sigma}$. The hybridization $V$ locally mixes $c$ and $f$ electrons. The local Coulomb interaction $U$ acts only between the $f$ electrons. The chemical potential of the $f$ electrons is set to $E_f = -U/2$, which guarantees a half-filled system with a gap in the single-particle spectrum at the Fermi energy. Throughout this paper, we use $t = V = 1$.

## 2.2 Method

To find the ground state of the Hamiltonian, Eq. (1), in the thermodynamic limit, $L \to \infty$, we use the *variational uniform matrix product states* (VUMPS) formalism [42–45]. For all calculations in the paper, we exploit the spin-SU(2) symmetry and the charge-U(1) symmetry of the model [46, 47].

The computation of dynamic correlation functions at $T = 0$ is carried out in real space by assembling a finite section of this translationally invariant solution and acting with the corresponding local operator. This local excitation is then propagated in real-time until a cutoff time, $t_{\max}$. This approach eliminates finite-size effects as long as the excitation does not reach the boundaries of the heterogeneous section.

Finite temperatures $T > 0$ are incorporated in a standard way by endowing each physical site with an "ancilla site" that acts as a thermal bath, thereby doubling the system size [48–51]. For $T > 0$, we use open boundary conditions, prepare the initial state at $\beta = 1/T = 0$, and then propagate it in imaginary time to $|\beta\rangle = \exp(-\beta H/2)|\beta = 0\rangle$. The partition function is thus $Z(\beta) = \langle \beta|\beta\rangle$ and observables are obtained via $\langle \cdot \rangle_\beta = Z^{-1}(\beta)\langle \beta|\cdot|\beta\rangle$. The corresponding operators only act on physical sites so that a trace over the bath is implicitly included in this average.

A central thermodynamic property is the specific heat per site, which is obtained from the internal energy $E(\beta) = \langle H \rangle_\beta$ as:

$$c = \frac{1}{L}\frac{\partial E(\beta)}{\partial T} = \frac{1}{L}\left[ \langle H^2 \rangle_\beta - \langle H \rangle_\beta^2 \right]. \tag{2}$$

Similarly, we compute the susceptibility in an SU(2)-invariant fashion via

$$\chi = \frac{\beta}{L}\left[ \langle \vec{S}_{\text{tot}}^2 \rangle_\beta - \langle \vec{S}_{\text{tot}} \rangle_\beta^2 \right] = \frac{\beta}{L} \langle \vec{S}_{\text{tot}}^2 \rangle_\beta \,, \tag{3}$$

where $\vec{S}_{\text{tot}} = \sum_j \vec{S}_j$ is the total spin operator and $\vec{S}_j = (S_j^x, S_j^y, S_j^z)$ is the local spin operator. In both cases, these formulas are directly evaluated by representing the corresponding observables as matrix-product operators using lossless compression.

## 2.3 Dynamical Correlation Functions

We analyze several dynamical correlation functions, $A_{\mu\nu}(\omega, k)$, which are calculated by Fourier transform from real-space and real-time correlation functions as [45, 52, 53]

$$A_{\mu\nu}(\omega, k) = \int_0^\infty dt \, e^{i\omega t} \sum_{mn} A_{m\mu, n\nu}(t) e^{-ik(m-n)L_c} \,, \tag{4}$$

where $m$ and $n$ are lattice-site indices, and the Greek indices $\mu, \nu$ label the two electron species $c$ and $f$. $L_c = 2$ is the length of the unit cell that is the consequence of putting the $c$ and $f$ sites on a "flattened" chain geometry. Regarding the real-time Fourier transform, we multiply the real-time data by a windowing function [52], $W(t) = \exp(-4t/t_{max})$, to avoid Gibbs artifacts due to the finite propagation time. This window function affects the width of the features in the spectra but not their positions. Typical real-time correlation functions are shown in the appendix B.

In particular, we analyze the single-particle Green's function $G^{1p}$, defined as

$$G^{1p}_{m\mu, n\nu}(t) = -i\Theta(t)\left[ \langle e^{-iHt} a^\dagger_{m\mu\sigma} e^{iHt} a_{n\nu\sigma} + e^{iHt} a_{m\mu\sigma} e^{-iHt} a^\dagger_{n\nu\sigma} \rangle_\beta \right], \tag{5}$$

where $a^{(\dagger)}_{m\mu\sigma}$ corresponds to $c^{(\dagger)}_{m\sigma}$ for $\mu = c$ and to $f^{(\dagger)}_{m\sigma}$ for $\mu = f$.

Furthermore, we will look at the CSF and the SSF, defined as

$$S_{m\mu,n\nu}(t) = -i\Theta(t)\langle e^{iHt}\vec{S}_{m\mu}e^{-iHt}\cdot\vec{S}_{n\nu}\rangle_\beta, \tag{6}$$

$$C_{m\mu,n\nu}(t) = -i\Theta(t)\langle e^{iHt}N_{m\mu}e^{-iHt}N_{n\nu}\rangle_\beta, \tag{7}$$

where $\vec{S}_{m\mu}$ is the spin operator as before; and $N_{mc} = n_m^c - \langle n_m^c \rangle$ ($N_{mf} = n_m^f - \langle n_m^f \rangle$) is the charge operator for the $c$ ($f$) electrons on lattice site $m$. SSF and CSF describe two-particle excitations in the system. In the absence of correlations, the Fourier-transformed CSF can be expressed by the convolution ("bubble diagram") of the single-particle Green's function as

$$C_{\mu\nu}^0(k,\omega) = \int d\omega' \int dk' G_{\mu\nu}^{1p}(k+k',\omega')f(\omega')G_{\nu\mu}^{1p}(k',\omega+\omega')(1-f(\omega+\omega')), \tag{8}$$

where $f(\omega)$ is the Fermi function. Thus, these functions can be interpreted as single-particle excitations from occupied states to unoccupied ones. For the noninteracting model, CSF and SSF are identical. In the presence of correlations, they are different and given by the corresponding two-particle Green's functions.

## 2.4 Current-current correlation function

Besides analyzing the single-particle Green's functions, CSF, and SSF, we will look at the current-current correlation function describing transport. In particular, we will focus on the electric charge current and the heat current. The current operator for the charge current can be calculated as the time derivative of the polarization operator $X$:

$$J_C = \partial_t X = i\left[H, \sum_j j\left(n_j^c + n_j^f\right)\right]. \tag{9}$$

For the periodic Anderson model in 1D with local hybridization $V_{ij} = V\delta_{ij}$ and vanishing $f$-electron hopping, $t_f = 0$, the charge current is carried by the $c$ electrons only, so that the operator becomes:

$$J_C = -it\sum_j\left(c_{j\sigma}^\dagger c_{j+1,\sigma} - c_{j\sigma}^\dagger c_{j-1,\sigma}\right). \tag{10}$$

In the same way, we can calculate the operator of the heat current, which is identical to the energy current in the case that the Fermi energy is at $\omega = 0$. We start from

$$J_E = i\left[H, \sum_j jh_j\right], \tag{11}$$

where $h_j$ is the "Hamiltonian density", satisfying

$$H = \sum_j h_j. \tag{12}$$

For the 1D periodic Anderson model, this reads

$$
\begin{aligned}
h_j = &-\frac{t}{2}\sum_\sigma\left(c_{j\sigma}^\dagger c_{j+1,\sigma} + c_{j-1,\sigma}^\dagger c_{j\sigma} + c_{j\sigma}^\dagger c_{j-1,\sigma} + c_{j+1,\sigma}^\dagger c_{j\sigma}\right) \\
&+ V\sum_\sigma\left(c_{j\sigma}^\dagger f_{j\sigma} + f_{j\sigma}^\dagger c_{j\sigma}\right) + E_f\sum_\sigma n_{j\sigma}^f + Un_{j\uparrow}^f n_{j\downarrow}^f.
\end{aligned} \tag{13}
$$

When calculating the commutators, we find that for the given model, neither $E_f$ nor $U$ contribute to the heat current, which is again due to a local hybridization and vanishing $f$-electron hopping. We obtain the following result:

$$J_E = it^2 \sum_{j\sigma} \left( c_{j-1,\sigma}^\dagger c_{j+1,\sigma} - c_{j+1\sigma}^\dagger c_{j-1,\sigma} \right) + i\frac{Vt}{2} \sum_\sigma \left( c_{j\sigma}^\dagger f_{j+1,\sigma} + f_{j\sigma}^\dagger c_{j+1,\sigma} \right)$$
$$- i\frac{Vt}{2} \sum_\sigma \left( c_{j\sigma}^\dagger f_{j-1,\sigma} + f_{j\sigma}^\dagger c_{j-1,\sigma} \right). \tag{14}$$

Having defined the charge-current and heat-current operators, we can calculate the real part of the transport functions as [54]

$$\text{Re}(L_{11}(\omega)) = \frac{1-e^{-\beta\omega}}{\omega} \int_0^\infty dt \cos(\omega t) \, \text{Re}(\langle J_C(t) J_C \rangle_\beta), \tag{15}$$

$$\text{Re}(L_{12}(\omega)) = \frac{1-e^{-\beta\omega}}{\omega} \int_0^\infty dt \cos(\omega t) \, \text{Re}(\langle J_E(t) J_C \rangle_\beta), \tag{16}$$

$$\text{Re}(L_{22}(\omega)) = \frac{1-e^{-\beta\omega}}{\omega} \int_0^\infty dt \cos(\omega t) \, \text{Re}(\langle J_E(t) J_E \rangle_\beta). \tag{17}$$

The electric conductivity and thermal conductivity are then given as [54]

$$\sigma(\omega) = L_{11}(\omega), \tag{18}$$

$$\kappa(\omega) = \frac{1}{T} \left( L_{22}(\omega) - \frac{L_{12}(\omega)^2}{L_{11}(\omega)} \right). \tag{19}$$

Thus, the DC conductivities can be calculated as the integral over the time-dependent correlation functions.

# 3 Correlation functions

## 3.1 $T = 0$

We start our analysis by examining the ground state properties of the periodic Anderson model in 1D. In this section, we use variational uniform matrix product states (VUMPS) corresponding to results in the thermodynamic limit, and we use a propagation time of $t_{max} = 60$. To show the feasibility of our method, we display the results for $U = 0$ in Fig. 1, which can be calculated exactly by diagonalizing the Hamiltonian. We use $E_f = -U/2$, so that $\langle n_j^c \rangle = \langle n_j^f \rangle = 1$ holds

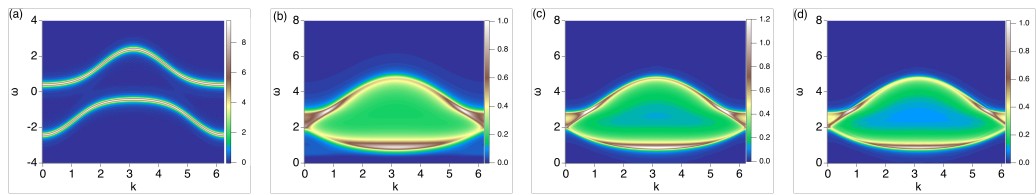

Figure 1: We show the single-particle spectrum $A^{1p}(\omega,k) = -\frac{1}{\pi}\text{Tr}\left\{\text{Im}(G_{\mu\nu}^{1p}(\omega,k))\right\}$ (a); the convolution of the single-particle Green's function, as written in Eq. (8), $-\frac{1}{\pi}\text{Tr}\left\{\text{Im}(C_{\mu\nu}^0(\omega,k))\right\}$ (b); the spectrum of the CSF, $-\frac{1}{\pi}\text{Tr}\left\{\text{Im}(C_{\mu\nu}(\omega,k))\right\}$ (d); and the spectrum of the SSF, $-\frac{1}{\pi}\text{Tr}\left\{\text{Im}(S(\omega,k))\right\}$ (d). All results are for $U = 0$ and $T = 0$.

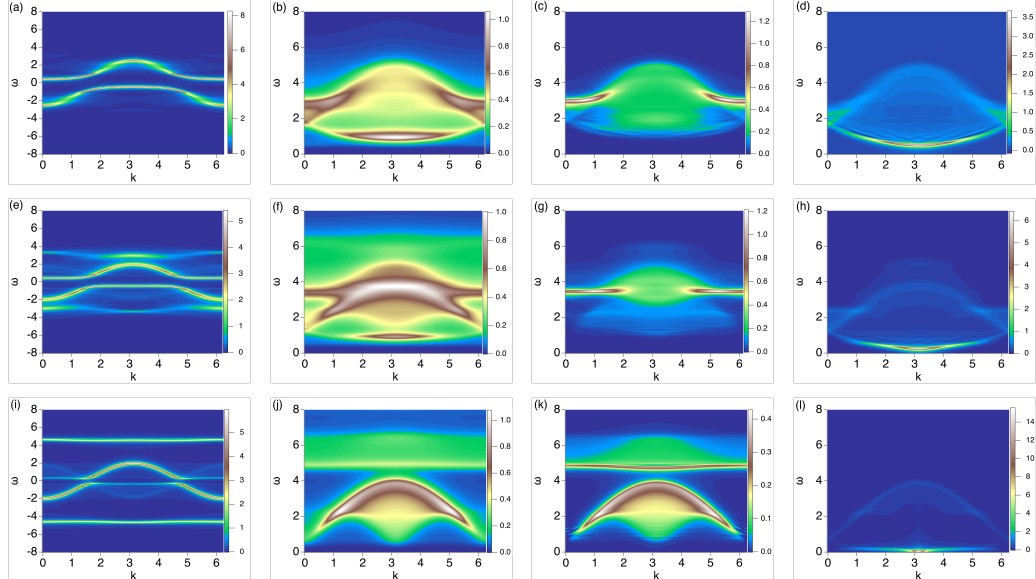

Figure 2: The same as in Fig. 1 but for the interacting system. The top panels (a-d) show results for $U = 2$, the middle panels (e-h) show results for $U = 4$, and the bottom panels (i-l) show results for $U = 8$. The left panels (a,e,i) show the single-particle spectral function for the corresponding interaction strengths. The second panels from the left (b,f,j) show the CSF as calculated from the convolution of the single-particle Green's function. The third panels (c,g,k) show the full CSF, as calculated by the VMPS. The fourth panels (d,h,l) show the SSF, as calculated by the VMPS.

for all lattice sites. These parameters correspond to a band insulator. We note that the VMPS method is based on entanglement properties of quantum systems so that the noninteracting (but still entangled) case, in fact, poses a similar challenge for the method as the interacting one.

The single-particle spectral function shown in Fig. 1(a) agrees with the result obtained by diagonalization of the noninteracting Hamiltonian. Clearly, it shows a gap at the Fermi energy, indicating the insulating ground state. As expected, the CSF, as shown in Fig. 1(c) is nearly identical to the SSF, Fig. 1(d). In the noninteracting model, charge and spin excitations correspond to electrons changing their energy. Furthermore, the convolution of the single-particle Green's function, as shown in Fig. 1(b), also agrees with the CSF and SSF. A slight difference in the broadening of structures is explained by the finite time (frequency) resolution of the correlation functions and bears no physical significance.

Next, we analyze excitations in the interacting model. Figure 2 shows the single-particle spectrum, the CSF, and the SSF for $U = 2$, $U = 4$, and $U = 8$. We depict the CSF calculated directly from the VMPS, including vertex corrections, and compare it to the CSF calculated by a convolution of the single-particle Green's functions neglecting vertex corrections, Eq. (8). This gives us the possibility to discuss the importance of vertex corrections in the Kondo insulator. Equation (8) can be interpreted as the amplitude of excitations in the single-particle spectral function from below to above the Fermi energy. Using this knowledge, we can relate structures in the CSF to excitations in the single-particle Green's function in Fig. 3.

The single-particle spectral functions, as shown in Fig. 2(a, e, i), remain gapped at the Fermi energy for all here considered interaction strengths. However, it is crucial to notice that the spectral weight of the $f$ electrons around the Fermi energy diminishes, which is easily observable in Fig. 2(i), where the spectral weight at $k = \pi$ in the band slightly below the Fermi

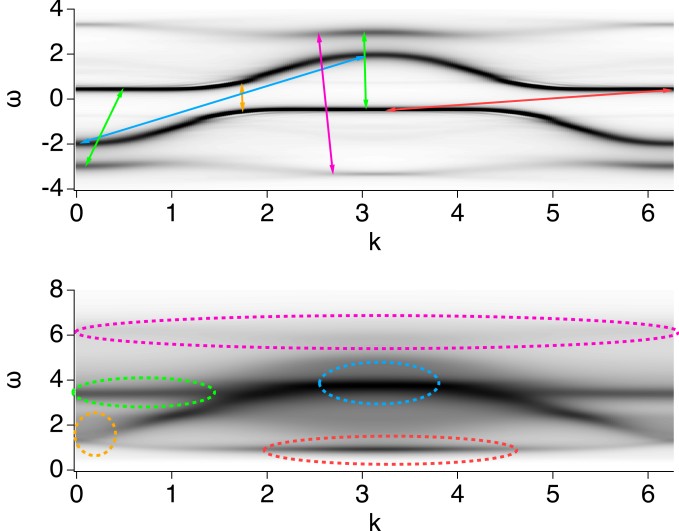

Figure 3: Single-particle Green's function (top) and convolution (bottom) of it for $U = 4$. Colored arrows denote possible excitations in the single-particle Green's function from below to above the Fermi energy. We use the same color in the convolution to explain the origin of the spectral weight in the CSF by single-particle excitations.

energy is clearly weaker than for $U = 0$. For strong interaction strengths, the spectral weight of the $f$ electrons is mainly found in the Hubbard bands at $\omega = \pm U/2$. This redistribution of spectral weight from the Fermi energy to the Hubbard bands with increasing interaction strength is a significant feature of the Kondo insulator, which also becomes relevant for two-particle spectral functions.

Comparing the CSF for different $U$, as shown in Fig. 2(c, g, k), with the convolution of the Green's function, as shown in Fig. 2(b, f, j), we find a qualitative agreement, in particular for weak and strong interaction strengths. The main difference is the distribution of spectral weight in the CSF, which is changed by the vertex corrections. Although the distribution of spectral weight is very different for intermediate interaction strengths, even for $U = 4$, all structures in the CSF can be found in the convolution. Thus, the CSF can be qualitatively understood by single-particle excitations described by the single-particle Green's function. For a better understanding of the excitations visible in the CSF, we show the single-particle Green's function and the convolution for $U = 4$ in Fig. 3 again. The colored arrows in the single-particle Green's function denote excitations from below to above the Fermi energy. We use the same colors in the convolution to demonstrate the origin of the spectral weight. It becomes clear that the broad continuum of particle excitations around $k = \pi$ is due to $c$-electron excitations. It is important to note that this continuum does not start at $\omega = 0$ but at a finite frequency which corresponds to the fact that we are analyzing an insulator. We furthermore understand that the flat band in the CSF, which is visible for $\omega > 3$ in Fig. 2, originates in excitations between the $f$-electron Hubbard band and $f$ electrons close to the Fermi energy (green color in Fig. 3). We note that the rather abrupt ending of this band-like structure can be understood by the momentum dependence of the spectral weight of the Hubbard bands. Contrary to our recent study of the Hubbard model, we do not find exceptional points in the CSF [55]. The excitations at low frequencies around $k = \pi$ (red color), which are pronounced in the convolution but less visible in the full CSF, originate in excitations between $f$ electrons close to the Fermi energy. Thus, with increasing interaction strength, when the spectral weight of the $f$ electrons close to the Fermi energy is diminished, also low-energy excitations in the CSF vanish.

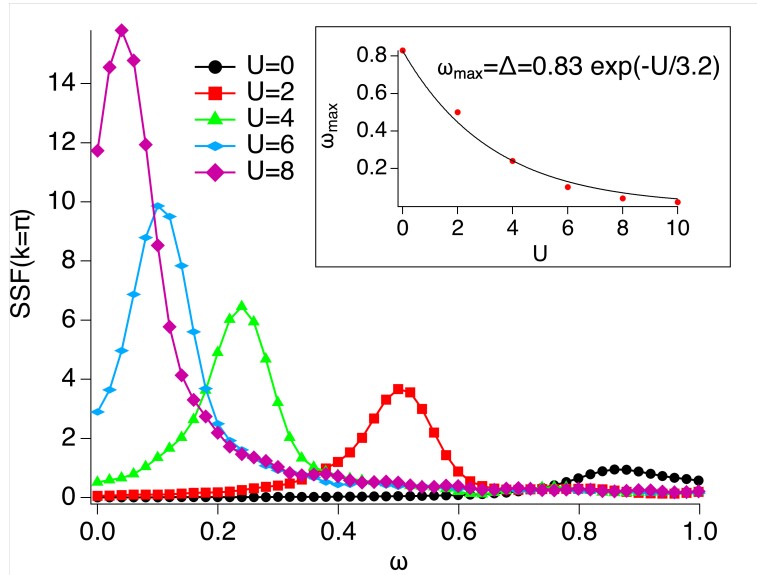

Figure 4: SSF spectrum at $k = \pi$ for different interaction strengths. The inset shows the frequency of the peak over the interaction strength.

Interestingly, for large interaction strengths, $U = 8$, the full CSF, and the convolution agree very well. The CSF is given by the continuum of $c$-electron excitations plus a flat band of $f$-electron excitations at high energies. Thus, vertex corrections do not play an essential role in the charge excitations at these interaction strengths.

Now, let us turn to the SSF, as shown in Fig. 2(d, h, l). While we see similar excitations as in the CSF, the most prominent feature is a flat band with a strong peak at $k = \pi$ at low energies. At $U = 0$, this band of spin excitations becomes the band of excitations seen in the convolution of the single-particle Green's function at $\omega \approx 1.6$ in Fig. 2(b). For $U = 0$, this band originates in excitations of $f$ electrons close to the Fermi energy. These excitations quickly diminish in the CSF and become pure spin excitations, whose energy decreases with increasing interaction strength. These spin excitations have much smaller excitation energy for large interaction strengths than the excitations seen in the CSF. Previous studies have predicted these excitations using the density matrix renormalization group [14–22] or variational Monte Carlo [23–27], focusing on the spin gap.

We stress here that our solution does not make any assumptions about the ground state of the Kondo insulator. As can be seen in Fig. 2, the energy of this excitation depends on the momentum, particularly for weak interactions. The lowest energy is reached for $k = \pi$. We show the SSF for different interaction strengths at $k = \pi$ in Fig. 4. We see that the energy of this excitation quickly decreases with increasing interaction strength. At the same time, the amplitude of the excitation strongly increases at $k = \pi$. The energy of this excitation thereby follows an exponential law as shown in the inset of Fig. 4. Thus, while charge excitations have a large gap, as shown above in the CSF, the spin gap becomes very small and follows a Kondo-temperature-like behavior for large interaction strengths. This difference can also be seen in the correlation lengths of spin-spin and charge-charge correlations, shown in appendix A.

In Fig. 5(a), we show this dispersion for different $U$ at $T = 0$. The dispersion has been extracted from the SSF spectrum by finding the maxima at small energies in the SSF. We show that the dispersion changes its behavior towards $k \to 0$ with increasing interaction strength. For small $U$ and, in particular, for $U = 0$, the energy of this excitation, i.e., the frequency, becomes large for $k \to 0$. For strong interactions, on the other hand, we observe that this

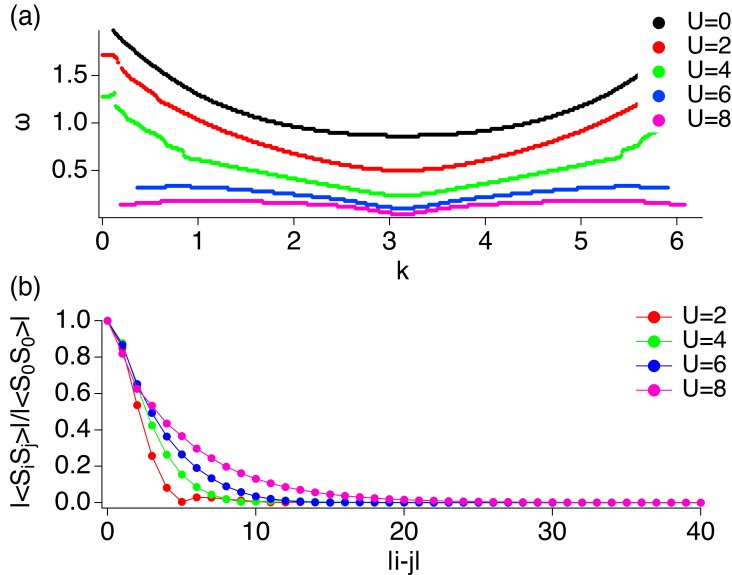

Figure 5: Energy-momentum dispersion of the energetically low-lying excitation in the SSF for different U at $T = 0$ (a). The bottom panel shows the strength of energetically low-lying correlations in real space (b).

excitation has low energy for all $k$. Furthermore, we note that most of the spectral weight of this spin excitation is located at $k = \pi$, as shown in Fig. 2. The dispersion of this spin excitation is confined to a small energy interval for strong interaction strengths. By calculating the Fourier transform of the momentum, we can get an image of the real-space extension of this spin excitation. We note that this excitation extends over a large energy interval for weak interaction strengths, corresponding to a traveling wave. We show the Fourier transform of the dispersion in Fig. 5(b). In particular, for small interaction strengths, the Fourier transform yields a very localized excitation. With increasing interaction strength, the extension of the excitation strongly increases. Many $f$ electron spins are involved in creating this low-energy spin excitation for strong interaction strengths. This energetically low-lying spin excitation, whose energy depends exponentially on the interaction strength, is the remainder of the Kondo effect in a Kondo insulator. In contrast to a Kondo impurity or a metallic Kondo lattice, the exponentially small energy scale cannot be found in the single-particle spectrum but only in the spin excitation spectrum of a 1D Kondo insulator. Furthermore, the Fourier transform of the momentum dispersion shows that the length scale of this excitation increases with increasing interaction. This also agrees with the picture of the Kondo effect, where the length scale of the Kondo singlet exponentially depends on the interaction strength.

## 3.2 Finite temperatures

Up to now, we have analyzed excitations at $T = 0$. A significant advantage of VMPS is that dynamical correlation functions can be calculated with high accuracy even at finite temperatures. We use a system size of $L = 60$ (physical) sites and a propagation time of $t_{max} = 20$. In Fig. 6, we show the single-particle spectrum ($A^{1P}(\omega)$), the CSF spectrum, and the SSF spectrum for $U = 6$ at $T = 2$, $T = 0.2$, and $T = 0.033$. As expected, at high temperatures, the $f$ electrons are localized, i.e., there are separated $c$-and $f$-electron bands visible in the single-particle spectrum at $T = 2$ (Fig. 6(a)). The $f$-electron spectral weight is mainly in the Hubbard bands at $\omega \approx \pm U/2$. The system at this temperature is in a metallic state. With decreasing the tempera-

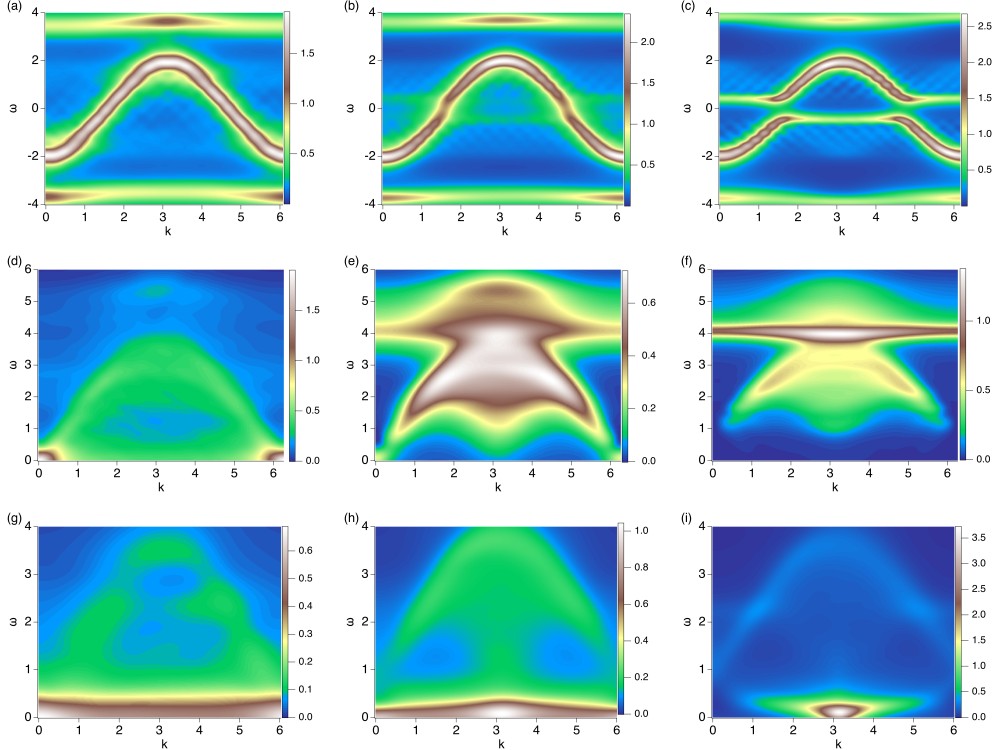

Figure 6: Single-particle spectral functions ($A^{1P}(\omega)$), the CSF spectrum, and the SSF spectrum for $U = 6$. The top panels show the $A^{1P}(\omega)$ for $T = 2$ (a), $T = 0.2$ (b), and $T = 0.033$ (c); The middle panels show the CSF spectrum for $T = 2$ (d), $T = 0.2$ (e), and $T = 0.033$ (f); and the bottom panels show the SSF spectrum for $T = 2$ (g), $T = 0.2$ (h), and $T = 0.033$ (i).

ture, $f$ electrons begin to hybridize with the $c$ electrons and form a gap at the Fermi energy, as visible in Fig. 6(b) and (c). Charge excitations, as shown in Fig. 6(d-f), have a strong peak at $k = 0$ at high temperatures corresponding to $c$ electron excitations around the Fermi energy. These ($k = 0$, $\omega = 0$)-excitations quickly vanish when decreasing the temperature. On the other hand, spin excitations in the SSF, shown in Fig. 6(g-i), are visible at high temperatures as a broad flat band at $\omega = 0$. Strongly correlated $f$ electrons are localized at this temperature and form a localized spin-1/2 object. With decreasing temperature, the spectral weight of this band accumulates at $k \approx \pi$, forming a large peak in the SSF at finite energy, $\omega > 0$. Thus, spin excitations, which consist of a single $f$ electron at high temperatures (flat band in momentum space), change to an extended object consisting of many atoms at low temperatures (a peak at $k = \pi$ in the SSF spectrum).

To further analyze the finite-temperature behavior, we compare the single-particle spectrum at $k = \pi/2$ for $U = 2$, $U = 4$, and $U = 8$, as shown in Fig. 7. At this momentum, the quasi-particle band of the unhybridized $c$ electrons intersects with the Fermi energy. In the spectrum for $U = 2$, Fig. 7(a), we see a gap (dip) at the Fermi energy for all temperatures. The $f$ and $c$ electrons are hybridized even at high temperatures. Decreasing the temperatures at this interaction strength shows only slight changes in the single-particle spectrum at the Fermi energy. For $U = 4$ and $U = 8$, the single-particle spectrum has a peak at high temperatures at the Fermi energy. When the temperature is decreased down to $T = 0.1$, the peak at the Fermi energy changes to a dip for $U = 4$. On the other hand, the peak remains nearly unchanged for $U = 8$ down to $T = 0.1$ and develops a dip only at $T < 0.1$. The different temperature behavior in the spectral function for $U = 4$ and $U = 8$ can be explained by different Kondo

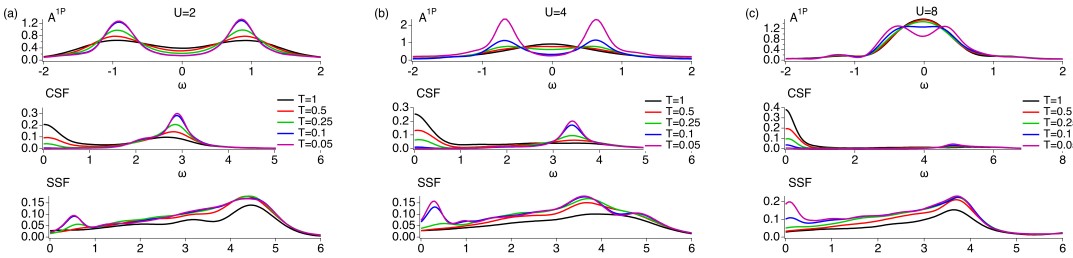

Figure 7: Single-particle spectral functions ($A^{1P}(\omega)$) at $k = \pi/2$, CSF spectrum at $k = 0$, and SSF spectrum at $k = \pi$ for U=2, U=4, and U=8 and different temperatures.

temperatures. We judge that the Kondo temperature for $U = 4$ is between $0.25 < T_K < 1$, while the Kondo temperature for $U = 8$ is between $0.025 < T_k < 0.1$. The different behaviors between weak interaction strengths, $U = 2$, and strong interaction strengths, $U = 4$ and $U = 8$, can be explained by the strength of the imaginary part in the self-energy of the single-particle Green's function. The imaginary part in the self-energy is large at strong interaction strengths and high temperatures, which leads to a separation of $c$ and $f$ electrons. Thus, the single-particle spectrum at the Fermi energy is given by the unhybridized $c$ electron band. When the temperature of the system is decreased at this interaction strength, the imaginary part of the self-energy also decreases, resulting in the appearance of an exceptional point at the Fermi energy and a hybridization gap for lower temperatures [56,57]. On the other hand, at weak interaction strengths, the imaginary part of the self-energy is not large enough to create exceptional points and separate $c$ and $f$ electrons even at large temperatures. Thus, the hybridization gap is present for all temperatures.

In Fig. 7, we also show the CSF and the SSF at $k = 0$ and $k = \pi$, respectively. In our analysis, we choose these momenta because the strongest changes occur here at the Fermi energy, $\omega = 0$, as shown in Fig. 6. We see that the CSF has a peak at $\omega = 0$ at high temperatures, which vanishes when the temperature is decreased. On the other hand, we see that the SSF develops a strong peak at low frequencies when the temperature is decreased. Most remarkable is, however, the difference in the temperature dependencies of the single-particle Green's function, the CSF, and the SSF. Particularly at large interaction strengths, $U = 8$, we see little changes in the spectral function and the SSF at $\omega = 0$ for temperatures $T > 0.25$. In the CSF ($U = 8$), on the other hand, the value at $\omega = 0$ decreases from 0.4 at $T = 1$ to 0.1 at $T = 0.25$. Low energy density-density correlations quickly vanish at high temperatures, while the single-particle spectral function and the SSF spectrum do not change. Furthermore, we see that the temperature range in which changes occur in the single-particle spectrum and the SSF spectrum strongly depends on the interaction strength and can occur at very low temperatures, while the changes in the CSF spectrum always occur at high temperatures.

Now, let us analyze the temperature dependence of the entropy per site, specific heat (Eq. (2)), and static spin susceptibility (Eq. (3)), which are shown in Fig. 8. The specific heat shows a two-peak structure. Using the entropy and spin susceptibility, we can understand the origin of these peaks. At high temperatures, the entropy takes the value $S(T = \infty) = \ln(16)$, corresponding to 16 possible Fock states per atom in the periodic Anderson model. This entropy decreases to a value slightly above $S \approx \ln(2)$, where the decrease in the entropy flattens. The formation of a plateau at $S \approx \ln(2)$ reveals the appearance of an effective unscreened spin-1/2 object at intermediate temperatures. The peak in the specific heat at $T \approx 1$ is related to this screening process. The entropy is further screened at even lower temperatures, forming a nondegenerate (spin-singlet) ground state visible in the specific heat as a second peak at low

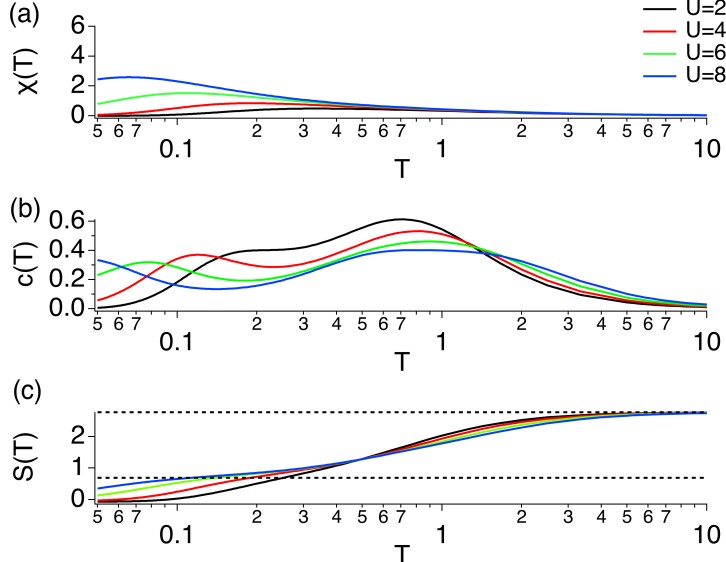

Figure 8: Spin susceptibility (a), specific heat (b), and entropy per site (c) for different temperatures and interaction strengths. Dashed lines in (c) correspond to ln(16) and ln(2).

temperatures. The temperature of the second peak in the specific heat thereby agrees with the energy scale set by the low-lying spin excitations seen in the SSF. Furthermore, the spin susceptibility, showing a strong increase when decreasing the temperature, further confirms an effective spin-1/2 state at intermediate temperatures. Only at low temperatures, corresponding to the low-temperature peak in the specific heat, the spin susceptibility forms a peak and finally vanishes at zero temperature. These results are in agreement with previous results in the 1D Kondo lattice model [20]. We always find a peak in the specific heat at low temperatures corresponding to the energy scale of the low-lying spin excitations well below the energy scale set by the gap in the single-particle Green's function. We note that peaks at low temperatures have been observed in Kondo insulators and are often interpreted as Schottky contributions originating, e.g., in the nuclear spin. Our study reveals that the Kondo screening of the localized $f$ electron spins can take place in a Kondo insulator at a much lower temperature than the opening of the gap resulting in a peak in the specific heat.

## 4 Transport properties

We now turn to the transport properties of this model. We analyze the electric charge and thermal transport in the 1D periodic Anderson model at finite temperatures. We calculate the time-dependent current-current correlation functions for heat and charge transport. The corresponding operators are given in Eqs. (10) and (14). For our comparison of the DC conductivity, we compare different systems consisting of $L = 80$, $L = 120$, and $L = 160$ sites. To calculate the time evolution, we separate forward and backward time propagation as described in Ref. [58]

$$|\Psi_f\rangle = \exp(-iHt/2)J_i|\Phi(\beta)\rangle, \tag{20}$$

$$|\Psi_b\rangle = \exp(iHt/2)J|\Phi(\beta)\rangle, \tag{21}$$

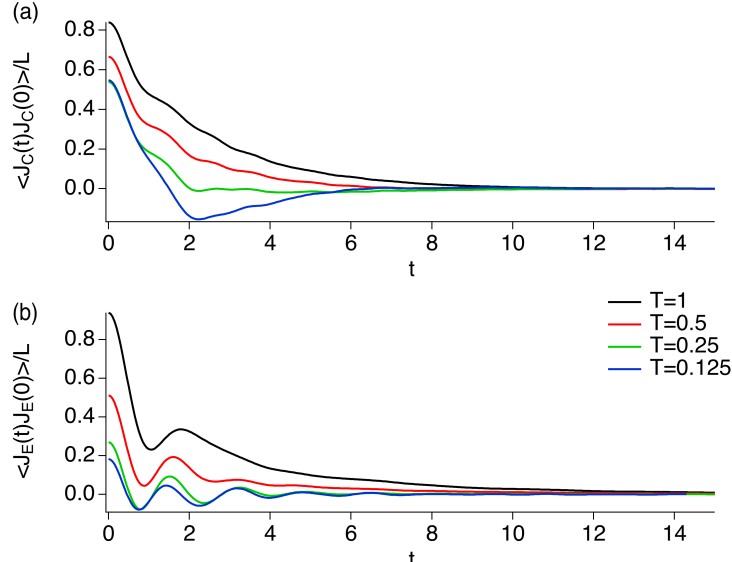

Figure 9: Charge (a) and heat (b) current-current correlation functions for $U = 4$ and different temperatures.

where $J_i$ is the current operator (charge or heat) at site $j$ in the middle of the chain, and $J$ is the full (sum over all sites) current operator (charge or heat). Because the system is homogeneous, we use the current operator of a single site for the forward propagation. $|\Phi(\beta)\rangle$ corresponds to the density matrix for a given $\beta$. The current-current correlation function is then given by the overlap, $\langle\Psi_b|\Psi_f\rangle$. During the time evolution, we increase the matrix size until it reaches the maximum matrix size. The maximum matrix size for the forward propagation is $m_f = 1000$, and the maximum matrix size for the backward propagation is $m_b = 2000$. We note that all truncation numbers, such as $m_b = 2000$, correspond to SU(2)-invariant states. For lower temperatures and longer chains, the maximal truncation is reached during this time evolution. In these cases, we continue the calculation using $m_f = 1000$ and $m_b = 2000$ and compare the current-current correlation functions for different values of $m$ and chain lengths to obtain a solution with high accuracy and get an estimate for the accuracy. A comparison of current-current correlation functions for different truncation levels is shown in the appendix C. We note that the calculation of the current-current correlation function at finite temperatures for these strongly correlated systems is a numerically complex task that reaches the limits of state-of-the-art computer systems. The results for the current-current correlation function at long times can thus not be seen as exact. However, after comparing different truncation levels, we can say that we are able to obtain results with sufficient accuracy for $U < 10$ and $T > 0.1$.

Typical heat- and charge-current correlation functions for $U = 4$ and different temperatures are shown in Fig. 9. All correlation functions quickly decrease for increasing time and finally vanish. This fast decrease in correlations helps obtain accurate DC conductivities because we only require to know the correlation functions on a timescale of $t < 15$. Besides this general decrease of the correlations with increasing time, we see additional oscillations appearing at low temperatures. These oscillations correspond to a finite conductivity at finite frequencies.

From the time-integral over the current-current correlation functions, we calculate the (DC) charge and thermal conductivities, Eqs. (18,19). Furthermore, we fit the conductivities by those of the noninteracting periodic Anderson models with renormalized hybridization strength $V_{\text{eff}}$, where $V_{\text{eff}}$ becomes a fitting parameter. To obtain the best fit, we compare noninteracting systems with different effective hybridization strengths and choose the system with

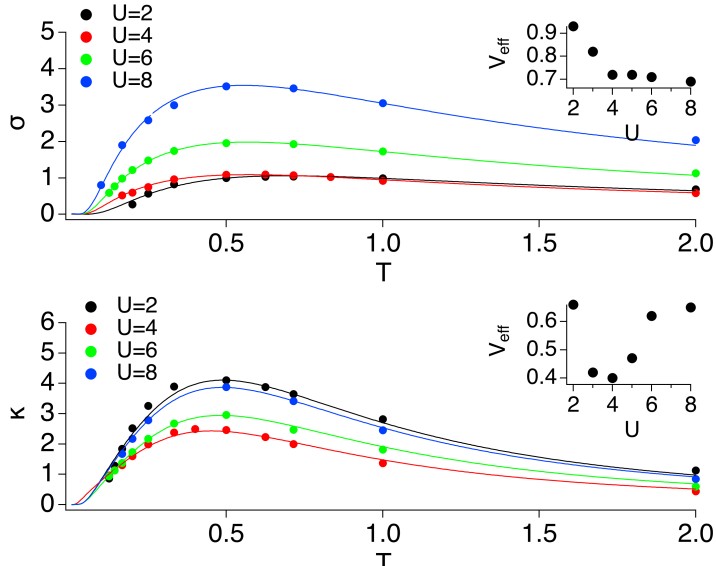

Figure 10: Charge Conductivities, $\sigma$, and thermal conductivities, $\kappa$, over temperature for different interaction strengths. Points denote the results obtained by the VMPS. Lines correspond to the best fit of the effective noninteracting system to the VMPS results. The used effective hybridization strengths are shown in the insets.

the smallest discrepancy to the conductivity of the interacting system. Independent of the hybridization strength, the noninteracting system is always gapped. Thus, this fitting yields information about whether the conductivity of the interacting system can be described by a gapped system and how large the gap in this effective system is. We perform this fitting for all interaction strengths separately for the charge and thermal conductivities. We fit the conductivities calculated by VMPS current-current correlation functions with conductivities calculated from a noninteracting periodic Anderson model, Eq. (1) ($U = 0$) using

$$\sigma = \int dk \int d\omega \left( -\frac{\partial f(\omega)}{\partial \omega} \right) \mathrm{Tr}(v A(\omega) v A(\omega)) \,, \tag{22}$$

$$\kappa = \frac{1}{T} \int dk \int d\omega \left( -\frac{\partial f(\omega)}{\partial \omega} \right) \omega^2 \mathrm{Tr}(v A(\omega) v A(\omega)) \,, \tag{23}$$

$$v = \frac{\partial H_0}{\partial k} \,, \tag{24}$$

$$A(\omega) = -\frac{1}{\pi} \mathrm{Im}(G^{1p}(k, \omega)) \,, \tag{25}$$

where $v$ is the velocity operator, and $f(\omega)$ the Fermi distribution function. We note that the transport function $L_{12}$ vanishes for the half-filled model for all temperatures. Thus, the thermal conductivity is given by $L_{22}$.

The results for the thermal and charge conductivities, including the fits to the effective noninteracting system, are shown for four different interaction strengths in Fig. 10. We see that both the charge and thermal conductivities for each interaction strength can be fitted well by a noninteracting system with effective hybridization strength, $V_{\mathrm{eff}}$. Thus, the charge and thermal conductivities, shown in Fig. 10, correspond to gapped systems. However, the hybridization strengths and, thus, the gap widths of the effective noninteracting systems are different for each interaction strength and also differ between the charge and the thermal conductivity.

The maximum of the charge conductivity over the temperature monotonically decreases with increasing interaction strength, as also demonstrated by the monotonic decrease of $V_{\text{eff}}$ in the inset of the charge conductivity. As expected, we see that with increasing interaction strength, the gap of the system becomes smaller. Remarkably, the maximum of the thermal conductivity generally lies at smaller temperatures than the maximum of the charge conductivity. The effective hybridization describing the thermal conductivity is always smaller for the thermal conductivity. Furthermore, we see that the maximum for $U = 4$ is at a lower temperature than that of $U = 6$ and $U = 8$, which is also confirmed by the smallest value of $V_{\text{eff}}$ for $U = 4$ in the inset of the thermal conductivity. We thus find a nonmonotonic behavior of the gap for the thermal conductivity.

Because the maximum of the thermal conductivity is generally at lower temperatures than the maximum of the charge conductivity, we find a region where the thermal conductivity is still increasing when the temperature is lowered, but the charge conductivity is decreasing. We must note, however, that the experimental observations of thermal metallic behavior in charge-insulating Kondo systems have been done at much lower temperatures. While our results show that two-particle correlations affect the charge and thermal conductivities differently, resulting in a region with high thermal and small charge conductivity, and may be a clue to understanding the intriguing experimental observations, our results cannot be directly used to explain the experimental findings.

A question remains about the nonmonotonic behavior and the minimum of $V_{\text{eff}}$ in the thermal conductivity for $U = 4$. While it is possible that the gap in the thermal conductivity in the weak coupling and strong coupling regions show different dependencies on the temperature, there is another possibility. We have shown above that there are two-particle excitations, such as spin excitations, whose energy depends exponentially on the interaction strength. It is possible that these excitations can carry heat but cannot carry charge. While the energy of the spin excitations, shown in Fig. 4, is $\omega \approx 0.24$ for $U = 4$, it is $\omega \approx 0.1$ for $U = 6$. It is possible that the discrepancy between the gap in the thermal conductivity and the charge conductivity is caused by these two-particle excitations. For $U = 4$, the energy gap of the quasiparticles, carrying charge and heat, and the quasiparticles carrying only heat is of the same order resulting only in a shift of the maximum of the thermal conductivity. On the other hand, for $U > 6$, the energy gap in the quasiparticle, carrying charge and heat, results in the maximum in the charge and heat conductivity at similar temperatures, while the two-particle excitations, carrying only heat, affect the thermal conductivity at much lower temperatures. This would result in discrepancies between the calculated thermal conductivity and the noninteracting fit. However, because we are currently only able to calculate conductivities for $T > 0.1$ with high accuracy, we cannot see these discrepancies at very low temperatures.

## 5 Conclusions

We have examined the 1D periodic Anderson model at half-filling as a prototypical model of a Kondo insulator using variational matrix product states. We have calculated dynamical correlation functions, such as the single-particle Green's functions, the CSF, and the SSF at $T = 0$ and $T > 0$. We have confirmed the existence of energetically low-lying spin excitations visible in the SSF, which form a flat band with a strong peak at $k = \pi$ for strong interaction strengths. We have found that the frequency of this peak exponentially approaches $\omega = 0$ with increasing interaction strength. Furthermore, by Fourier transform, we calculated the real-space extension of these spin excitations and found that the length scale of these spin excitations strongly increases with increasing interaction strength; many f-electron spins are involved at large interaction strengths. For strong interaction strengths, the gap in the single-particle Green's

function is much larger than the energy of these spin excitations. Thus, the exponential energy scale of the Kondo effect at which the $c$ electrons screen the $f$-electron spins is not visible in the single-particle Green's functions but is visible in the SSF. Furthermore, by examining the thermodynamic quantities, we have seen that these spin excitations are observable in the specific heat and static spin susceptibility. Peaks at low temperatures have been observed in Kondo insulators and are often interpreted as Schottky contributions originating, e.g., in the nuclear spin. Our study reveals that the Kondo screening of the localized $f$ electron spins can take place at a much lower temperature in a Kondo insulator than the opening of the gap resulting in a peak in the specific heat at very low temperatures.

Finally, we have studied the charge and thermal conductivity of this model at finite temperatures. We have found that gapped quasiparticles can explain both the charge and thermal conductivities. However, we found that the gap in the thermal conductivity is generally smaller than the gap in the charge conductivity; two-particle excitations affect thermal and charge conductivity differently. Furthermore, we have found that the gap in the thermal conductivity behaves nonmonotonically in the temperature region, in which we can accurately calculate the conductivity. This can be explained by energetically low-lying excitations, such as the confirmed spin excitations, that affect the thermal conductivity but do not affect the charge conductivity. While for weak interaction strengths, the effect of these excitations occurs at high enough temperatures to be detected by our calculations, at strong interactions, the effect occurs at too low temperatures. However, to confirm this hypothesis, novel and even more powerful approaches should be developed to access the ultra-low temperature regime.

# Acknowledgements

The computation in this work has been done using the facilities of the Supercomputer Center, the Institute for Solid State Physics, the University of Tokyo.

**Funding information** R.P. is supported by JSPS, KAKENHI Grant No. JP18K03511.

# A    Correlation length

We show in Fig. 11 the charge-charge correlation, $\langle N_j N_0 \rangle$, and the spin-spin correlation, $\langle \vec{S}_j \vec{S}_0 \rangle$, for different interaction strengths. We note that $c$ electron and $f$ electron correlations follow the same long-range exponential law. Thus, we only show the $f$ electron correlations. From the long-range behavior, we calculate the correlation length for the charge-charge and spin-spin correlations, which are shown in Fig. 11(c). We see that the spin-spin correlations are decreasing much more slowly than the charge-charge correlations, resulting in a longer spin correlation length, especially for large $U$. The spin correlation length increases exponentially, while the charge correlation length increases linearly (for the interaction strengths calculated). This agrees with the expectation that the length scale of Kondo correlations increases exponentially with $U$ and that these Kondo correlations are only seen in the spin correlation function in the Kondo insulator.

# B    Time-dependent correlation functions

The spectral functions shown in the main text are calculated by the Fourier transform of real-time and real-space correlation functions. In Fig. 12, we show typical time-dependent corre-

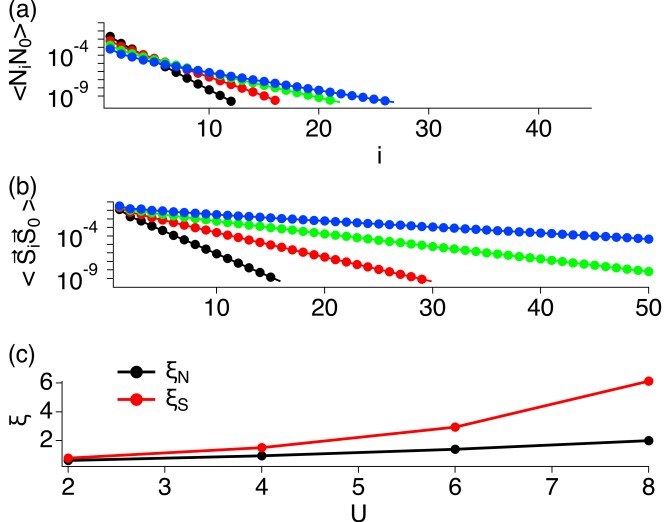

Figure 11: Charge-charge (a) and spin-spin correlations (b) and the corresponding correlation lengths (c) at $T = 0$. These correlation functions were calculated with a maximal bond dimension of 500. We only show correlations between the $f$-electrons but note that the $c$-electron correlations follow the same exponential law for long distances. The correlation length is calculated by fitting an exponential law as $\langle A_j A_0 \rangle = a \exp(-j/\xi_{N/\vec{S}})$, where the operator $A$ is either the charge density, $N$, or the spin, $\vec{S}$. The correlation length is given in units of the distance between two atoms, including a $c$ electron level and an $f$ electron level.

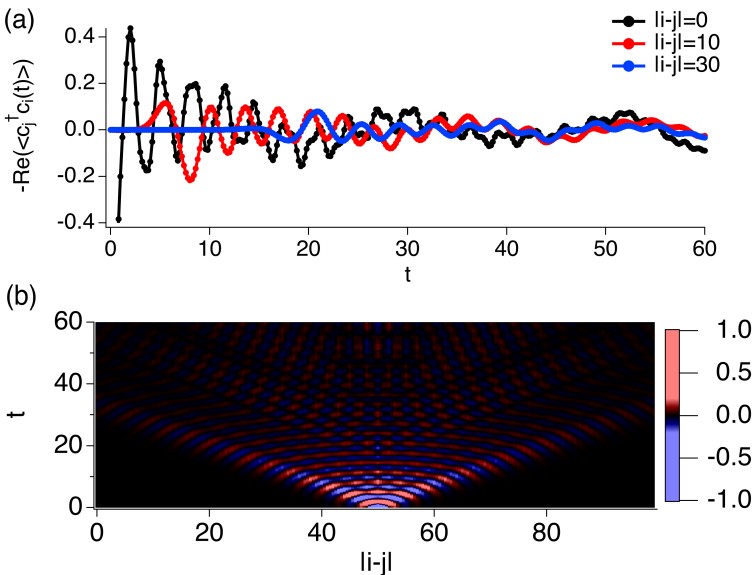

Figure 12: Time-dependent correlation functions $-\text{Re}\langle c_i^\dagger(t)c_j \rangle$ for different distances $|i-j|$. The parameters are $U = 8$ and $T = 0$.

lation functions, $-\text{Re}\langle c_i^\dagger(t)c_j \rangle$, for different distances $|i-j|$. At $t = 0$, the correlation function is strongly localized around the origin of the excitation. This excitation propagates through space with increasing time.

## C  Truncation effects

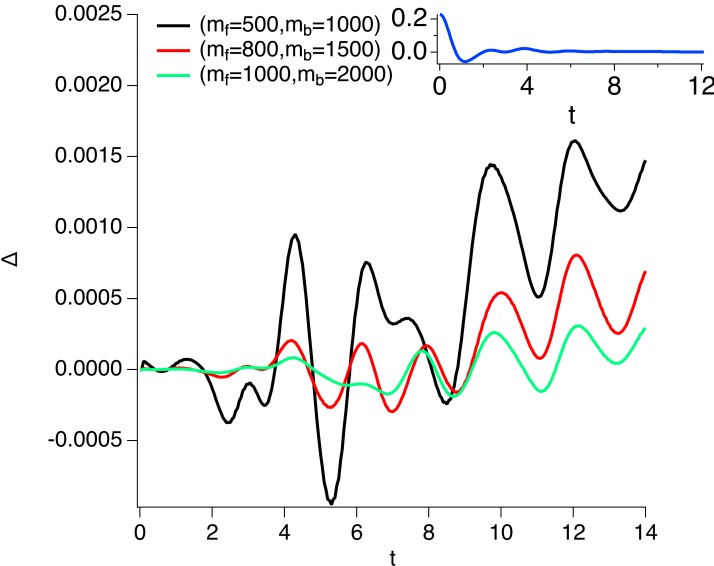

Figure 13: Differences of the heat current correlation function for $U = 4$ and $\beta = 4$ for different truncation levels. The inset shows the heat current correlation function for $(m_f = 1500, m_b = 2500)$.

To obtain an estimate of the accuracy of the calculated conductivities, we performed several calculations using different truncation levels. A typical result of such a comparison for $U = 4$ and $\beta = 4$ is shown in Fig. 13. The actual results in the main text are obtained for a truncation using $(m_f = 1000, m_b = 2000)$, corresponding to the truncation level during the forward and backward propagation. These bond dimensions are chosen in the calculations as large as possible to ensure high accuracy but small enough that the calculation can finish in a reasonable calculation time of several days. In Fig. 13, we show the differences between this truncation and lower truncation levels with $(m_f = 1500, m_b = 2500)$, which must be assumed closer to the exact result. We see that the difference between the $(m_f = 1500, m_b = 2500)$ result and the $(m_f = 1000, m_b = 2000)$ result is always smaller than 0.0005. Furthermore, the integrated value of the current-current correlation function is also smaller than 0.0005. To obtain the thermal conductivity, we need to multiply this value with $\beta^2$, resulting in an estimated absolute error of 0.008 for $U = 4$ and $\beta = 4$. We repeated similar calculations for $U = 8$. We find that the error increases with decreasing temperatures and increasing interaction strengths. Furthermore, the factor $\beta^2$ additionally amplifies the error at small temperatures. Thus, we estimate that we can calculate conductivities with sufficient accuracy for $T > 0.1$.

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
