# Peer review of "Low-energy excitations and transport functions of the one-dimensional Kondo insulator"

_SciPost Physics, doi:SciPost Phys. 14, 166 (2023)_

## Round 1 · Referee Report · Anonymous · 2023-3-6

Strengths

1- Numerically exact variational matrix product states (VMPS) method yields
ground state properties representative of the thermodynamic limit.

2-Excitation spectra/correlation functions are calculated directly on the
real frequency/time axis.

3-Provides evidence for charge-neutral heat carriers in a Kondo insulator and
insight into their physical nature.

Weaknesses

1-The VMPS method is fully controlled only in a limited temperature range above the experimental observations of thermal metallic behavior in charge-insulating Kondo systems.

Report

The manuscript presents a thorough study of the momentum-resolved single-particle spectral function, spin and charge structure factors, and thermodynamic properties (Section 3) supplemented by the analysis of the charge and thermal conductivity (Section 4) of the one-dimensional periodic Anderson model at half filling.

The results of Section 3 nicely illustrate the key physics of the model, i.e., a separation of spin and charge energy scales with increasing Hubbard interaction strength which highlights the difference with respect to a conventional band insulator. However, these results do not contribute anything qualitatively new and can be found in the previous studies using for example a finite-T DMRG method.

What nevertheless argues in favor of publication in SciPost Physics is Section 4
devoted to the analysis of the charge and thermal conductivity at finite temperatures. Specifically, the authors find a temperature region where the charge conductivity of the Kondo insulator is already decreasing while the heat conductivity is still increasing.

This is an important novel result suggesting that experimentally observed thermal metallic behavior in charge-insulating Kondo system can arise solely as a result of strong correlations. Within this scenario, the heat transport is carried by low-energy spin excitations while the charge transport is blocked at temperatures smaller than the charge gap.

These findings provide a simple alternative to theories of charge-neutral fermions invoking topological effects, exotic quasiparticles, or phonons and shall stimulate follow-up work.

Requested changes

1-What is the actual value of the hybridization amplitude V used in the
simulations?

---

## Round 1 · Referee Report · Anonymous · 2023-3-6

Strengths

1- Detailed analysis of ground-state and thermodynamics properties of one-dimensional Kondo insulator.
2- Computation of finite-temperature charge and heat transport properties

Weaknesses

1- The finite-temperature parameter range is limited to moderate or high temperatures

Report

The authors investigate the finite-temperature properties of the one-dimensional (1d) Kondo insulator using matrix product states (MPS). First, they are able to reproduce known results at T=0 (spectral functions) as well as thermodynamics at finite T using state-of-the-art numerical techniques. But more interestingly, by computing time-dependent correlation functions, they can obtain the charge and thermal conductivities. It is known for a long-time that in such systems, there are various gaps: charge gap, single-particle gap, spin gap, which can be quite different. As a result, the heat and charge transport behave quite differently too, which could explain recent experiments.

By a careful comparison of exact 2-particle calculations (for charge or spin spectral functions), it is remarkable that qualitative features are found using a simple convolution of the 1-particle Green's function, i.e. without vertex corrections.

I find the paper well written and the results interesting. They are obtained using state-of-the-art numerical techniques. Hence, I recommend it for publication. I do have questions and suggestions though, that could help to improve the presentation:

* Is there any understanding why vertex corrections are negligible for strong coupling ? This could be useful for other techniques which often neglect them.

* In order to perform Fourier transform in time, did you use some trick to avoid artefacts due to the finite tmax ?It would be nice to see (in supplemental) some typical time-dependent correlations at T=0.

* To compute the specific heat, it is written that an MPO form was used for H^2. Is it exact or compressed ? Would it be easier to compute it from the energy e(T) ?

* It is mentioned that different bond dimensions are used for backward/forward evolution, probably due to the local/global nature of the operator. How were the numbers chosen ? In one Appendix, there is a benchmark plot but it would be useful to see how these two parameters are chosen.

* Why are the results not directly comparable to experiments ? Of course materials are 3d, but what about the typical temperature scales ?

Requested changes

1- It would be useful to plot the local density of states N(w) for various T in order to see the appearance of the gap and the emergence of Kondo physics.
2- What is the finite-temperature correlation length for T>0.1 ? Is it the limitation for the numerical technique ?
3- I would not call VMPS as "numerically exact" since usually MPS methods are not guaranteed to converge to the absolute minimum (finding the optimal MPS is NP hard). Of course they do work extremely well for simple models.

---

## Round 2 · Author Response

Report 1
We thank the referee for carefully reading our manuscript and recommending it for publication. Below we answer all questions.
* * *
* Is there any understanding why vertex corrections are negligible for strong coupling ? This could be useful for other techniques which often neglect them.
* * *
[Answer]
We do not know the full answer to this very interesting question.
However, it is important to stress that the spin-spin correlation function does not include the low-energy excitations without the vertex corrections. Thus, vertex corrections are not generally unimportant at strong interaction strengths!
On the other hand, we found that the bubble diagram reproduces well the charge-charge correlations without vertex corrections at strong interactions. Unfortunately, we do not fully understand when vertex corrections are important and when they are unimportant in the presence of correlations. It depends on the kind of correlation function that is studied.
* * *
* In order to perform Fourier transform in time, did you use some trick to avoid artefacts due to the finite tmax ?It would be nice to see (in supplemental) some typical time-dependent correlations at T=0.
* * *
[Answer]
As described in Phys. Rev. Lett. 93 076401, we use a windowing function that is multiplied with the real-time data to avoid Gibbs artifacts due to a finite propagation time.
In the revised manuscript, we mention this as
“Regarding the real-time Fourier transform, we multiply the real-time data by a windowing function, $W(t) = exp(-4*t/t_{max})$, to avoid Gibbs artifacts due to the finite propagation time. This window function affects the width of the features in the spectra, but not their positions.”
Furthermore, we show examples of time-dependent correlation functions (of the single-particle Green’s function) in a new appendix of the revised manuscript.
* * *
* To compute the specific heat, it is written that an MPO form was used for H^2. Is it exact or compressed ? Would it be easier to compute it from the energy e(T) ?
* * *
[Answer]
In our calculations, H^2 is expressed exactly (with lossless compression). Obtaining H^2 generally poses no problem for short-ranged Hamiltonians with small MPO bond dimensions. Computing the specific heat by a numerical derivative of e(T) would be equally possible. However, it seems to us that using H^2 is more accurate and thus preferable whenever H^2 can be obtained.
We mention this in the revised manuscript.
* * *
* It is mentioned that different bond dimensions are used for backward/forward evolution, probably due to the local/global nature of the operator. How were the numbers chosen ? In one Appendix, there is a benchmark plot but it would be useful to see how these two parameters are chosen.
* * *
[Answer]
As written in the manuscript, we split the calculation of the current-current correlation functions by using a part of the current operator in the middle of the chain and calculating the correlation of this with the full current operator.
Thus, one propagation uses only a part of the current operator, while the other propagation uses the full current operator. This is the reason why we have chosen different bond dimensions for the backward and forward propagation.
Otherwise, we tried to use a large bond dimension (as large as possible), still ensuring that the calculation finishes in a reasonable time (several days) and does not consume too much memory (<200GB). After testing several different bond dimensions, which are shown in the appendix, we decided to use m_f=1000 and m_b=2000.
We mention this in the appendix of the revised manuscript as
“These bond dimensions are chosen in the calculations as large as possible to ensure high accuracy but small enough that the calculation can finish in a reasonable calculation time of several days.”
* * *
* Why are the results not directly comparable to experiments ? Of course materials are 3d, but what about the typical temperature scales ?
* * *
[Answer]
A direct comparison to experiments is difficult not only because of the dimension but also because of the band structure of the material.
In our calculations, the results strongly depend on the interaction strength. When comparing to real materials, on the other hand, we will need to take into account the full band structure.
For example, the resistivity of the topological Kondo insulator SmB6 starts to increase at around 30K. This change in resistivity corresponds to the initial decrease of the conductivity in our calculations at T=0.5 – 0.8 (depending on the interaction strength). Regarding this material, our calculations would thus be valid for T>5K. However, it is important to note that SmB6 is expected to be strongly correlated (large value of U) and that SmB6 has strong valence fluctuations (f electrons are not half-filled), while we only performed calculations for half-filled f electrons.
Thus, we believe that comparison to real materials is very difficult due to the band structure and the dimension.
Requested changes
* * *
1- It would be useful to plot the local density of states N(w) for various T in order to see the appearance of the gap and the emergence of Kondo physics.
* * *
[Answer]
We note that the density of states is shown in Fig. 7 of the manuscript. While our calculations show the gap formation for U=2 and U=4, U=8 has a Kondo temperature T<0.1. Thus, for U=8, we do not see the formation of a full gap in the finite T calculations.
* * *
2- What is the finite-temperature correlation length for T>0.1? Is it the limitation for the numerical technique ?
* * *
[Answer]
This is an interesting question.
We have calculated the charge-charge and spin-spin correlations at T=0, which are shown in a new appendix of the revised manuscript. We choose to calculate these correlations at T=0 because the correlation length generally decreases with increasing temperature. Thus, we believe T=0 shows the largest correlation length for a certain interaction strength.
In our calculations, we see that charge-charge and spin-spin correlations follow a different exponential law at long distances. Furthermore, we see that the correlation length of spin-spin correlations increases exponentially with the interaction strength while the charge-charge correlation length behaves (rather) linearly. This agrees well with our dynamical correlation functions, where we also see a Kondo scale in the spin-spin correlations, which is absent in the charge-charge correlations.
Furthermore, we see that the spin-spin correlation length for U=8 is around 6. For this interaction strength, our systems are much larger than the correlation length. Thus, there are no limitations due to the correlation length for U<10. The limitations in our calculations arise due to the entropy growth during the time evolution, which depends on the interaction strength and the temperature.
* * *
3- I would not call VMPS as "numerically exact" since usually MPS methods are not guaranteed to converge to the absolute minimum (finding the optimal MPS is NP hard). Of course they do work extremely well for simple models.
* * *
[Answer]
We thank the referee for pointing this out.
We have changed the expression “numerically exact” to “highly accurate.”
* * *
* * *
* * *
Report 2
* * *
We thank the referee for carefully reading our manuscript and recommending it for publication. Below we answer all the raised questions.
* * *
Requested changes
1-What is the actual value of the hybridization amplitude V used in the
simulations?
* * *
[Answer]
We thank the referee for asking this question. We indeed missed giving the actual number. All calculations are done for V=1.
We now include these values in the revised manuscript.

---

## Round 2 · List of Changes

• We shortly comment on how the Fourier transform of the real-time data is performed. Furthermore, we show a few examples of real-time and real-space correlation functions in the appendix.
• We mention that H^2 is uncompressed.
• We mention how we chose the bond dimension for the forward and backward propagation.
• We have calculated the correlation length for several interaction strengths and show these in the appendix.
• We have changed “numerically exact” to “highly accurate.”
• We have included the values of V and t in the model section.

---

## Editorial Decision

published